# Pleasantness Ratings of Musical Dyads in Cochlear Implant Users

**DOI:** 10.3390/brainsci12010033

**Published:** 2021-12-28

**Authors:** Andres Camarena, Grace Manchala, Julianne Papadopoulos, Samantha R. O’Connell, Raymond L. Goldsworthy

**Affiliations:** 1Auditory Research Center, Caruso Department of Otolaryngology, Keck School of Medicine, University of Southern California, Los Angeles, CA 90033, USA; andresc@usc.edu (A.C.); gmanchal@usc.edu (G.M.); jmpapado@usc.edu (J.P.); sroconne@usc.edu (S.R.O.); 2Thornton School of Music, University of Southern California, Los Angeles, CA 90089, USA

**Keywords:** auditory neuroscience, cochlear implant, hearing loss, music, pitch discrimination, plasticity, musical sophistication

## Abstract

Cochlear implants have been used to restore hearing to more than half a million people around the world. The restored hearing allows most recipients to understand spoken speech without relying on visual cues. While speech comprehension in quiet is generally high for recipients, many complain about the sound of music. The present study examines consonance and dissonance perception in nine cochlear implant users and eight people with no known hearing loss. Participants completed web-based assessments to characterize low-level psychophysical sensitivities to modulation and pitch, as well as higher-level measures of musical pleasantness and speech comprehension in background noise. The underlying hypothesis is that sensitivity to modulation and pitch, in addition to higher levels of musical sophistication, relate to higher-level measures of music and speech perception. This hypothesis tested true with strong correlations observed between measures of modulation and pitch with measures of consonance ratings and speech recognition. Additionally, the cochlear implant users who were the most sensitive to modulations and pitch, and who had higher musical sophistication scores, had similar pleasantness ratings as those with no known hearing loss. The implication is that better coding and focused rehabilitation for modulation and pitch sensitivity will broadly improve perception of music and speech for cochlear implant users.

## 1. Introduction

Music is a powerful tool used to express and elicit emotion. It can be a deeply personal source of enjoyment. Hearing loss, however, can dampen or distort incoming sound, and thus greatly reduce music appreciation. Cochlear implants (CIs) restore hearing in people with sensorineural hearing loss and largely rehabilitate speech understanding without the need for visual cues. Despite having generally high levels of speech comprehension, music appreciation in CI users is largely diminished compared to their normal-hearing peers [1,2]. In particular, CI users struggle with facets of musical listening including pitch perception, instrument identification, and melody recognition [3,4,5]. This dampened enjoyment of music has come to be expected by both clinicians and prospective CI recipients and can be a major factor in determining whether a person goes forward with implantation [6]. Therefore, further investigation is warranted to determine the aspects of hearing that significantly impact how music is appreciated in the hard-of-hearing community.

The perceptual deficits that CI users face are largely caused by technological limits of the implanted electrodes. While the healthy auditory nerve contains around 30,000 fibers, cochlear implants use an array of no more than 22 electrodes. This limited range of electrodes along with the spread of electrical current reduces the resolution needed to resolve harmonics [7,8,9]. Likewise, temporal cues for pitch are dampened by how sound processing for cochlear implants converts sound into electrical stimulation. In healthy hearing, when multiple harmonics interact within the cochlea of the inner ear, they produce a temporal beating at the fundamental frequency. This beating produces deep modulations in the auditory nerve response, which is a clear temporal cue for pitch perception. In contrast to the cochlear filters of healthy hearing, cochlear implants use narrow filters with minimal overlap [10,11]. Discrete filters are used to allocate narrow frequency bands to the limited number of implanted electrodes. The consequence, however, is that fewer harmonics interact within a filter, which results in shallow modulation of the neural response—timing cues relevant to the perception of pitch.

Poor representation of place and timing cues for pitch and timbre in CIs has a marked effect on music appreciation for CI users [3,4,5,12,13]. Pitch is an important aspect of music listening—it is the sensation of hearing a single sound from a complete range of sounds and is a building block to create musical melodies. The pitch of a harmonic signal is most saliently derived from place-of-excitation cues associated with the fundamental frequency and lower harmonics of the fundamental [14,15,16,17,18,19]. For higher harmonics, cochlear filters become increasingly broad, and harmonic components no longer provide discernable place cues for pitch. However, interactions between harmonic components within a single filter result in a neural response with a periodic fluctuation in amplitude. The repetition rate of this amplitude modulation is a temporal cue for pitch that can be perceived at least up to around 500 Hz in normal-hearing adults but is often lower in those with hearing impairment [20,21,22,23,24,25,26].

Timbre is also a vital component to enjoying music. It is described by the American Standards Association (1960) as “that attribute of sensation in terms of which a listener can judge those two sounds having the same loudness and pitch are dissimilar”, and is often associated with the character or brightness of a sound. While timbre is often defined by what it is not (e.g., that it is not loudness or pitch), it can be clearly described by several acoustic features. For instance, by extending the attack time of a trumpet, it becomes qualitatively like a violin; yet the two instruments can still be distinguished by their unique spectral content even if they were playing the same note. The timbre of the harmonic signal is comprised of three acoustic features that drive timbre perception, including the temporal envelope, the spectral envelope, and spectral flux [27,28,29,30]. The independence between pitch and timbre was demonstrated in a psychophysical experiment by Plomp and Steeneken (1971) where they concluded that timbre has a perceptual correlate of spectral excitation along the basilar membrane [31]. Together, pitch and timbre are both needed to provide the scaffolding that makes the perception of voiced speech and musical notes an enjoyable experience.

Oftentimes, songs are not constructed out of single notes but composed of *chords*: musical structures formed by the simultaneous presentation of two or more harmonic sounds. Music’s perceived pleasantness is driven by how well the sources harmonize. The degree of harmony, however, varies with the interval distance between the combined harmonics. A single harmonic series will have overtones spaced at integer multiples of the fundamental. When played simultaneously with another harmonic series, the components of each series may fuse in pleasant consonance. In contrast, the overtones produced at more awkward intervals may sound harsh and dissonant. For example, complexes with a fundamental ratio of 1:1, 1:2, or 2:3 tend to sound consonant, whereas ratios of 8:9, 8:15, or 32:45 are often considered dissonant [32,33,34,35,36]. When described in musical notation, unison, octave, and perfect fifth are considered consonant, while major second, major seventh, and tritone are dissonant. In the present study, pleasantness ratings of two-note chords referred to as “dyads” is examined in detail.

One factor that may impact music enjoyment for CI users may be their perception of consonance and dissonance. While CI users can rank or rate stimuli based on dissonance, performance is often poorer or less pronounced than in normal hearing listeners [37,38,39]. While their absolute ratings were often lower than normal hearing listeners, Spitzer and colleagues (2008) demonstrated that elements of the profile of pleasantness ratings across intervals was shared for CI users and normal-hearing listeners [40]. This consistency across groups, however, was mostly in the perception of dissonance with CI users only displaying a mild sensitivity to consonance at an octave interval. Analysis of the modeled output of the CI suggests that spectral cues did not contribute strongly to ratings of harmonic intervals but were instead likely driven by temporal envelope cues. Therefore, further study is required to characterize the psychophysical cues that lend themselves to the perceived pleasantness of musical harmony.

Studies also demonstrate that previous levels of musical experience, including active music listening and engagement, can influence performance on musical tasks. For example, LoPresto (2015)’s work on consonance and dissonance demonstrates that normal hearing, musically trained participants were more likely than non-musically trained participants to indicate that they disliked the sound of dissonant intervals in comparison to consonant intervals [41]. Music training studies with adult CI users also provide evidence that attentive music listening and engagement can lead to improved performance on frequency change detection and speech in noise identification [42]. We, therefore, predict that higher levels of musical experience among both normal hearing listeners and CI users will be positively correlated with music and speech perception.

The purpose of this study was to characterize the pleasantness profile of CI users across harmonic intervals and to determine the aspects of modulation and pitch perception that influence the perception of consonance and dissonance. Furthermore, this study also investigates how levels of musical sophistication influence CI users’ performance in these tasks. Pleasantness ratings were obtained from participants with no known hearing loss and from CI users for harmonic intervals spanning an octave. We hypothesized that sensitivity to temporal pitch cues is a driving factor in pleasantness ratings. Specifically, we predicted that CI users with pleasantness profiles that are most like normal-hearing listeners and those with higher levels of musical sophistication would be those most sensitive to amplitude modulations.

## 2. Methods

### 2.1. Participants

Nine total CI users (*R* = 36–83 years old, *M* = 65.5 years, *SD* = 13.7 years, females = 5) and eight individuals with no known hearing loss (*R* = 25–49 years old, *M* = 32.8 years, *SD* = 9.6 years, females = 3) took part in this experiment. Seven CI participants used Cochlear Corporation implants (Cochlear Americas, Lone Tree, CO, USA), one used an Advanced Bionics implant (Sonova, Los Angeles, CA, USA), and one used a Med-El implant (Med-El, Innsbruck, Austria). Complete CI participant information is provided in Table 1. Participants gave informed consent and were paid $15/hour for their participation. The experimental protocol was approved by the University of Southern California Institutional Review Board.

### 2.2. Materials and Procedure

People with no known hearing loss and CI users took part in an online listening experiment designed to characterize pleasantness ratings of musical dyads—pairs of musical notes presented simultaneously. All testing was done through *TeamHearing*: a free web-based software platform developed by our lab at USC for Aural Rehabilitation and Assessment (www.teamhearing.org, accessed on 20 December 2021). The *TeamHearing* web application includes a range of speech and pitch perception tests created to measure various aspects of hearing including musical judgements, psychophysical discriminations, and speech reception in various environments and noise conditions. The specific *TeamHearing* measures used for this study are described in the follow sections.

*TeamHearing* assessments were accessed on a personal computer, personal tablet, or a mobile device. People with no known hearing loss used headphones to complete the task. CI users were asked to complete the task in a method that was most comfortable for them, either by listening to the task through speakers or receiving sound input directly to their processor via Bluetooth or through a Mini Microphone device (Cochlear Americas, Lone Tree, CO, USA). Calibration of sound levels were conducted using loudness adjustments and detection thresholds for pure tones. Participants completed five assessments: modulation detection, fundamental frequency discrimination, consonance identification, pleasantness ratings for musical dyads, and speech reception thresholds on a sentence completion task in multi-talker background noise. Total testing time was two to three hours. A permalink for this experiment can be found at https://www.teamhearing.org/82, accessed on 20 December 2021.

### 2.3. Calibration

Before completing the assessments, participants completed procedures to characterize relative loudness levels. First, participants adjusted a 1 kHz pure tone to be “soft”, “medium soft”, “medium”, and “medium loud”. Second, pure tone detection thresholds were measured for octave steps between 125 and 8000 Hz. Stimuli were 400 ms sinusoids with 20 ms raised-cosine attack and release ramps. At the beginning of a measurement run, participants set the stimulus volume to be “soft but audible”. Detection thresholds were then measured using a three-interval, three-alternative, forced-choice procedure in which two of the intervals contained silence and one interval contained the gain-adjusted tone. Participants were instructed via on-screen instructions to select the interval that contained the tone. The starting gain was set by the participant and thereafter reduced by 2 dB after correct responses and increased by 6 dB after incorrect responses. A run continued until three mistakes were made and the average of the last four reversals was taken as the detection threshold. This procedure converges to 75% detection accuracy [43].

### 2.4. Modulation Detection

Modulation detection was measured for modulation frequencies near 10 and 110 Hz. These modulation frequencies were chosen as representative of a roughness cue relevant to harmonic distortion (10 Hz) and as representative of one of the relevant fundamental frequencies being examined for pleasantness ratings (110 Hz). Modulation detection was measured using a three-interval, three-alternative, forced-choice procedure where two of the intervals contained standard stimuli without modulation and one of the intervals was modulated with adaptively controlled modulation depth. The standard stimuli were 1 kHz pure tones that were 400 ms in duration with 20 ms raised-cosine attack and release ramps. The target stimulus was identically defined except being amplitude modulated. The initial modulation depth was set to 100%. The modulation depth was decreased by a factor of $\sqrt[{3}]{2}$ following correct answers and was increased by a factor of two following mistakes. This adaptive logic converges to 75% detection accuracy [43]. A measurement run ended after the participant made four mistakes and the average of the last four reversals was taken as the modulation detection thresholds. Each of the two modulation frequencies tested (10, 110 Hz) was measured with three repetitions with conditions presented in random order. Correct answer feedback was provided on all trials for this and all subsequent procedures except for pleasantness ratings (as there is no correct answer for that procedure).

### 2.5. Fundamental Frequency Discrimination

Fundamental frequency discrimination was measured for fundamental frequencies near 110, 220, and 440 Hz. These fundamental frequencies were chosen as representative of the typical range of spoken speech and as indicative of the range over which discrimination typically deteriorates for CI users. Discrimination was measured using a two-interval, two-alternative, forced-choice procedure for which participants were asked which interval was higher in pitch. The stimuli were complex tones constructed in the frequency domain by summing all harmonics from the fundamental to 2 kHz with a low pass filtering function. The form of the low pass filtering function was:$$gain=\{1\;if\;f<f_{e}\left(0.1-(f-f_{e}\;\right){\;}^{2})\;otherwise\;$$
where gain is the gain expressed as a linear multiplier applied to each harmonic component, f is the frequency of the component, and $f_{e}$ is the edge frequency of the passband, which was set as 1 kHz for the low pass filter. Note, as thus defined, the low-pass filter gain is zero above 2 kHz. Each measurement run began with a fundamental frequency difference of 100% (an octave). This difference was adaptively controlled and reduced by a factor of $\sqrt[{3}]{2}$ after correct responses and increased by a factor of two after incorrect responses. For each trial, the precise fundamental frequency tested was roved with values selected from a quarter-octave range uniformly distributed and geometrically centered on the nominal condition frequency. Relative to the roved value, the standard fundamental frequency was lowered, and the target raised by $\sqrt{1+\Delta /100}$. The gain of the standard and target were roved by 6 dB based on a uniform distribution centered on the participant’s comfortable listening level. A run ended when the participant made four incorrect responses and the average frequency difference of the last four reversals was taken as the discrimination threshold.

### 2.6. Consonance Identification

To test consonance identification, participants were asked to categorize dyads as either consonant or dissonant. Four-note dyads were examined including two dyads typically labeled as consonant (i.e., perfect fifth, octave) and two dyads typically labeled as dissonant (i.e., tritone, major seventh). Consonance identification was measured for root notes near 110, 220, and 440 Hz with each dyad type measured with ten trials each. All musical stimuli were generated using MuseScore 3 composition and notation software (https://musescore.org/en, accessed on 20 December 2021; Musescore BVBA, Belgium). Stimuli were grand piano notes with a duration of three seconds. Before commencing a measurement run, participants were first provided with several examples of consonant and dissonant dyads. During piloting of the pleasantness ratings procedures, it was noted that some CI users who had extensive musical experience could clearly hear the difference between dyads that are typically labeled as consonant (e.g., perfect fifth, octave) and those typically labeled dissonant (e.g., tritone, major seventh), though they were reluctant to assign the terms “pleasant” or “unpleasant” to this distinction.

### 2.7. Pleasantness Ratings

The same musical note stimuli described for consonance identification were used for collecting pleasantness ratings for all participants. Musical dyads were formed by combining two of the rendered piano notes with dyadic combinations including thirteen pairings of every note combination with semitone spacing ranging from unison (i.e., combining a note with itself) to an octave (i.e., combining a note with a note one octave higher). Dyads were organized into experimental conditions with pleasantness ratings collected for pairings near 110, 220, and 440 Hz. Musical dyads were presented one at a time. Participants were asked to rate the pleasantness of the dyad on a Likert scale from 0 to 6 with 0 labeled as “dissonant or unpleasant”, 3 as “neutral”, and 6 as “consonant or pleasant”. For a measurement run, each of the thirteen dyadic pairings (each semitone spacing from unison to octave, inclusive) were presented twice. A total of nine measurement runs were made including three repetitions of each of the three note ranges (110, 220, 440 Hz).

### 2.8. Speech Reception in Multi-Talker Background Noise

Speech reception thresholds were measured for a sentence completion task using speech materials from the Speech Perception in Noise Test (SPIN) corpus in the presence of multi-talker background noise [44]. The user interface presented twenty-five different word options, and participants were asked to choose the word that ended the last spoken sentence. The modified SPIN corpus contains sentence materials that include both high and low amounts of contextual information. Only the materials with low context information were used in the present study, since we are mainly concerned with the availability of low-level perceptual cues as opposed to cognitive factors. Speech reception thresholds were measured using an adaptive procedure. The initial signal to noise ratio between the spoken sentence and background noise was set to 12 dB and was decreased by 2 dB after correct responses and increased by 6 dB after incorrect responses. The procedure continued until the participants made four incorrect responses and the average of the last four reversals was taken as the reception threshold. This adaptive rule converges to 75% identification accuracy for the speech reception [43].

### 2.9. The Goldsmith Musical Sophistication Index

Musical experience was measured using the Goldsmith Musical Sophistication Index Self-Report Inventory (MSI), a 39-item psychometric instrument used to quantify the amount of musical engagement, skill, and behavior of an individual [45]. The questions on this assessment are grouped into five subscales: active engagement, perceptual abilities, musical training, singing abilities, and emotion. Questions under the active engagement category consider instances of deliberate interaction with music (i.e., “I listen attentively to music for *X* hours per day”). The perceptual abilities category includes questions about music listening skills (e.g., “I can tell when people sing or play out of tune”). Musical training questions inquire about individuals’ formal and non-formal music practice experiences (“I engaged in regular daily practice of a musical instrument including voice for *X* years”). Singing abilities questions inquire about individuals’ singing skills and activities (e.g., “After hearing a new song two or three times I can usually sing it by myself”). Questions under the emotion category reflect on instances of active emotional responses to music (e.g., “I sometimes choose music that can trigger shivers down my spine”). These topics together consider an individual’s holistic musical ability, including instances of formal and non-formal music training and engagement. The composite score of these subscales makes up an individual’s general musical sophistication score. All items, except those assessing musical training, are scored on a seven-point Likert scale with choices that range from “completely disagree” to “completely agree” [45].

### 2.10. Data Analysis

Data processing and statistical analyses were performed in MATLAB 2021a programming environment (MathWorks, Inc., Natick, MA, USA). Results from each test were analyzed using a 2 × 3 mixed analysis of variance (ANOVA) with a between-subject factor of group (CI versus those with no known hearing loss) and a within-subject factor of measurement repetition (three repetitions per test). Effect size was calculated using Cohen’s *d*. Post-hoc Bonferroni adjustments were performed for significant main effects [46]. Pearson’s bivariate correlations were calculated to investigate relationships between average scores on perceptual tests and musical sophistication measures.

## 3. Results

### 3.1. Calibration Procedures

Figure 1 compares loudness settings and pure tone detection thresholds for both participant groups. The difference in average detection thresholds between groups was significant ($F_{1,21}=19.2,\;p<0.001$) with CI users setting the average software volume higher (38.2±15.2 dB) compared to those with no known hearing loss (8.3±16.4 dB). Importantly, these thresholds are measured relative to the system volume that participants adjust their computers to for the at-home listening procedures. These results are not indicative of absolute detection thresholds, but they show that when participants adjust their computer and listening device settings to be comfortable, CI users have elevated detection thresholds. The effect of frequency was significant ($F_{1,21}=21.8,\;p<0.001$) as was the interaction between frequency and participant group ($F_{6,21}=3.2,\;p=0.005)$. The interaction effect is evidenced by CI users having particularly elevated thresholds for the lowest and highest frequencies tested.

### 3.2. Modulation Detection

Figure 2 shows modulation detection thresholds for 10 and 110 Hz modulation frequencies. Participants with no known hearing loss were more sensitive to modulations than the CI users ($F_{1,20}=17.2,\;p<0.001$). Modulation frequency affected sensitivity ($F_{1,20}=12.2,\;p=0.002$), and there was a significant interaction between modulation frequency and participant group ($F_{1,20}=16.7,\;p<0.001$). For those with no known hearing loss, modulation detection improved from 11.2% at 10 Hz to 4.5% at 110 Hz (d=2.8, p<0.001); for CI users, detection slightly worsened from 18.7 to 21.0% (d=0.14, p=0.63). Neither repetition nor the interaction between participant group and repetition was significant (p>0.1, for both comparisons) indicating that significant main effects were not influenced by an effect of learning.

### 3.3. Fundamental Frequency Discrimination

Figure 3 shows fundamental frequency discrimination thresholds measured near fundamental frequencies of 110, 220, and 440 Hz. Participants with no known hearing loss had better discrimination than CI users ($F_{1,21}=57.0,\;p<0.001$). Fundamental frequency affected sensitivity ($F_{2,21}=7.6,\;p=0.002$), and there was a strong interaction between fundamental frequency and participant group ($F_{2,21}=4.2,\;p=0.02$). For those with no known hearing loss, discrimination thresholds were around 0.5% with little variation across fundamental frequencies (p>0.1, for all comparisons). In contrast, for CI users, discrimination worsened from 5.5% at 110 Hz to 14.0% at 220 Hz ( d=0.74, p=0.01), then further worsened to 40.3% at 440 Hz (d=0.71, p=0.008). Neither repetition nor the interaction between participant group and repetition was significant (p>0.1, for both comparisons) indicating that significant main effects were not influenced by an effect of learning.

### 3.4. Consonance Identification

Figure 4 shows identification accuracy for consonant (unison, perfect fifth, and octave) and dissonant (major second, tritone, and major seventh) dyads. Participants with no known hearing loss had better identification than CI users ($F_{1,21}=11.2,\;p=0.003$). Neither fundamental frequency ($F_{2,21}=0.3,\;p=0.79$), nor the interaction between fundamental frequency and participant group ($F_{2,42}=0.1,\;p=0.94$) were significant. This contrasts with fundamental frequency discrimination. Neither repetition nor the interaction between participant group and repetition was significant (p>0.1, for both comparisons) indicating that significant main effects were not influenced by an effect of learning.

### 3.5. Pleasantness Ratings

Figure 5 shows pleasantness ratings for musical dyads ranging from unison to an octave in semitone increments. Averaged across all conditions, CI users rated dyads as less pleasant with an average rating of 2.86 compared to 3.46 for those with no known hearing loss ($F_{1,21}=6.5,\;p=0.02,\;d=0.37$). Importantly, the interaction between hearing group and dyadic interval was significant ($F_{12,252}=7.8,\;p<0.001$), indicating the profile differences in ratings. These underlying differences can broadly be seen in that CI users have flatter use of the ratings scale, but a more detailed profile analysis is considered in the subsequent paragraph. Overall consonance ratings of both groups were similar in that unison, perfect fourth, perfect fifth, and octave were consistently rated as more pleasant, while minor second, tritone, and major seventh were consistently rated as less pleasant. Additionally, and interestingly, a main effect of note range was observed for both groups. The average consonance rating across groups and intervals was higher for ascending root notes $\left(F_{2,21}=9.1,\;p<0.001\right)$. Grand averages of consonance ratings were 2.72, 3.12, and 3.35 for root notes near 110, 220, and 440 Hz, respectively.

Further analysis of the similarities between pleasantness ratings were conducted by calculating the correlation between individual ratings with the average ratings from the group with no known hearing loss. Figure 6 shows the correlations for each note range. For the individuals within the group with no known hearing loss, the correlations are high since this represents correlations of individual ratings trend with its own group average. These correlations indicate the consistency within the group. In contrast, the CI users exhibited a much greater variability with some participants having pleasantness ratings within the group range for those with no known hearing loss, while other participants exhibited no or even negative correlation. Thus, some CI users have near normal pleasantness ratings for two-note chords, while others flat or even opposing ratings.

### 3.6. Speech Reception in Multi-Talker Background Noise

Figure 7 shows speech reception thresholds for participants with no known hearing loss and for the CI users. Participants with no known hearing loss had better speech recognition with average thresholds of −11.0 dB compared to CI users with average thresholds of 9.2 dB (d=3.2, p<0.001). Thus, the difference between group averages was more than 20 dB. Neither repetition nor the interaction between participant group with repetition was significant (p>0.1, for both comparisons), indicating that significant main effects were not influenced by an effect of learning.

### 3.7. Correlation Analyses

Correlation analyses were conducted to consider relationships between average results across procedures. Specifically, for each procedure, measures were averaged across repetitions and conditions to yield a single value for each participant. These participant averages were then used to calculate the correlation between results across procedures. For pleasantness ratings, the correlation between individual pleasantness ratings and the group average for participants with no known hearing loss was used as the procedural result. Table 2 summarizes the correlations across procedures. All correlations were significant when considering the entire participant pool (p<0.001). To characterize the extent to which these strong correlations were driven by a group effect, separate correlation analyses were conducted for the two participant groups. For the group with no known hearing loss, significant correlations were found between modulation detection and speech reception thresholds, between fundamental frequency discrimination and both consonance identification and pleasantness ratings, and between consonance identification and pleasantness ratings. For the CI users, all correlations were generally strong with all associated *p*-values less than 0.1 and most less than 0.05. In summary, strong correlations were observed between measures with the strength of correlation generally persisting even when considering the participant groups separately.

As an example of specific correlations, Figure 8 compares performance on fundamental frequency discrimination, consonance identification, pleasantness ratings, and speech reception with modulation detection. Participants who had better modulation detection generally performed better or as well on the other procedures.

A final correlation analysis was conducted to compare MSI with performance on modulation detection, fundamental frequency discrimination, consonance identification, pleasantness ratings, and speech reception in noise. The composite general musical sophistication scores were used for correlations in data analysis. Table 3 shows that for normal hearing and CI users together, there is a significant correlation between MSI scores and all perceptual measures. The strong correlations between MSI and perceptual measures are generally preserved in the within-group correlations with a few exceptions. For those with no known hearing loss, the correlations between MSI and modulation detection and with speech reception in noise were not significant. For cochlear implant users, the correlation between MSI and speech reception in noise did not reach significance, but all other correlations with MSI were significant.

The perceptual measures are plotted against MSI for comparison in Figure 9. The clear trend is for better performance with higher MSI composite scores. The relationship is precise and well described by a linear relationship for modulation detection, fundamental frequency discrimination, consonance identification, and pleasantness ratings profile. The relationship is less precise and did not reach significance for within-group comparisons for speech reception in noise. 

## 4. Discussion

The hypothesis tested by this experiment is that low-level sensitivity to modulations and to pitch change are predictive of higher-level measures of consonance perception. We also predicted that higher levels of musical sophistication would be positively correlated with performance on music and speech perception tasks. The first hypothesis was supported by evidence of strong correlations between measures of modulation detection and fundamental frequency discrimination, along with higher-level measures such as consonance identification and speech reception in background noise. The second hypothesis was partially supported by strong correlations between musical sophistication scores and performance on fundamental frequency discrimination thresholds, consonance identification, and pleasantness ratings for both the no known hearing loss group and CI users. MSI scores for both groups were not significantly correlated with modulation detection or speech reception thresholds. Discussion is focused on the significance of these trends and how they relate to other aspects of hearing, such as audibility, pitch resolution, and speech comprehension in challenging environments.

The present experiment was in part motivated by a study by Spitzer and colleagues (2008), who examined pleasantness ratings in people who had a cochlear implant in one ear and normal hearing in the other [40]. In that study, the authors found that pleasantness ratings were generally flat across musical dyads when listening with the implanted ear. The authors noted that there were similarities in the pleasantness ratings between the implanted and normal-hearing ear; for example, participants tended to rate minor second and major seventh intervals as relatively dissonant in both ears. However, pleasantness ratings were generally flat as heard through the cochlear implant. The authors speculated that access to consonance perception provided by cochlear implants is likely mediated by modulation sensitivity, though they did not test this hypothesis explicitly.

In the present study, the relationships between modulation and pitch sensitivities with pleasantness ratings was explicitly considered. The results indicate that both low-level measures of modulation and pitch sensitivity are well correlated to consonance identification. Even when considering the CI users in isolation, the correlation between pitch discrimination with consonance identification and with pleasantness ratings profile was exceptionally strong. This evidence indicates that consonance perception amongst CI users is a broadly varying dimension of hearing, with the implant users who are most sensitive to modulations and pitch changes having the best access to consonance perception.

Results demonstrate that musical sophistication level is another factor that is strongly correlated with CI users’ perception of consonance. This is supported by previous works by LoPresto and Firestone which conclude that increased music training and engagement can lead to improved consonance identification and pitch discrimination [41,47]. In contrast to previous work, our results did not demonstrate in CI users a connection between music sophistication and speech reception in noise. Similarly, music training levels were not significantly correlated with low-level measures of modulation sensitivity as originally predicted. Together, these findings point to music sophistication as one of several factors that influence the perception of consonance.

A further contribution of the present study is the precise characterization of how strongly the pleasantness ratings of CI users can align with those with no known hearing loss. Previous studies have not considered how the general shape of pleasantness ratings as a function of dyadic interval compares for cochlear implant users. In the present study, the correlation between pleasantness ratings for individual CI users with a template from those with no known hearing loss clearly indicates that CI users can have normal pleasantness ratings. More specifically, levels of musical sophistication for CI users and those with no known hearing loss were significantly correlated with pleasantness ratings. This indicates that music training plays a similar role in CI users as in normal hearing populations when it comes to pleasantness identification. However, some CI users have distinctly abnormal ratings with negative correlation to those with no known hearing loss, suggesting a possible reversal in which dyadic intervals sound pleasant and which sound unpleasant.

Worth noting is that a strong correlation was also observed between pitch resolution and speech reception in multi-talker babble. The presumed mediating mechanism is that CI users who are more sensitive to pitch change can use this access to pitch to attend to target speech in the presence of competing talkers. While that mediating mechanism was not explicitly tested in the present study, the strong correlations support the conjecture. However, it is also possible that the best performing CI users are high performing on both pitch and speech tasks without the pitch mechanism necessarily facilitating speech recognition. Further evidence of the association, though not a causative relationship, was provided in previous work from our laboratory [48,49]. Returning to the present study, the strong correlations observed between pleasantness ratings profiles with speech reception provide further evidence of the association between musical and speech domains [50,51,52,53,54]. We presume that these relationships are partly driven by low-level access to psychophysical cues for modulation sensitivity and pitch resolution, though causality has not been established.

The present study demonstrates the importance of sensitivity to modulation and pitch detection in combination with music training in order to enhance consonance and dissonance identification abilities among CI users. With the ability to discriminate between consonant and dissonant sounds in music comes the potential to identify sounds that are more pleasant to an individual’s music listening [41]. In addition to these findings, there are certain limitations to consider. The limited number of participants and their individual differences should be considered. For example, some participants had years of musical training experience and had a greater understanding of consonance and the expected pleasantness of various musical intervals. Additionally, years of experience using cochlear implants, implant layout, and streaming method were not controlled among participants. While analyses conducted did attempt to limit the effect to which these individual differences could have an effect in the calculated threshold and correlation coefficients, further studies are needed to understand said differences. Additionally, it should be noted that the only aspect of music pleasantness measured in this study was for harmony at specific intervals —music perception in CI users in general is affected by various other factors such as simultaneous presentation of musical instruments and voices. The extent to which all factors play a role in perception should be carefully considered when analyzing temporal cues in sound processors.

## Figures and Tables

**Figure 1 brainsci-12-00033-f001:**
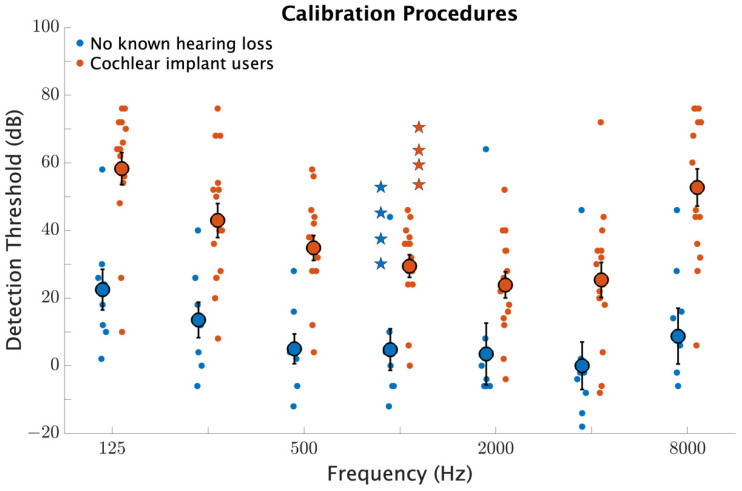
Comparison of loudness settings and detection thresholds for participants with no known hearing loss and for CI users. Detection thresholds and loudness settings are plotted in decibels relative to the maximum output volume of the testing device (computer or tablet) with 100 dB corresponding to the maximum output. Smaller circles indicate individual results and larger circles indicate group averages with error bars indicating standard errors of the mean. Stars indicate group averages for loudness levels corresponding to “soft”, “medium soft”, “medium”, and “medium loud”.

**Figure 2 brainsci-12-00033-f002:**
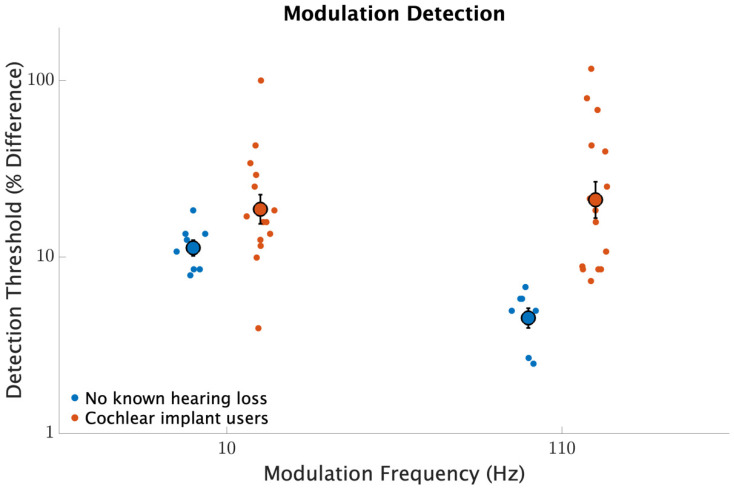
Modulation detection threshold as a percent difference for modulation frequencies of 10 and 110 Hz. The smaller circles represent individual detection thresholds. The larger circles with error bars represent across participant averages with error bars indicating standard errors of the means.

**Figure 3 brainsci-12-00033-f003:**
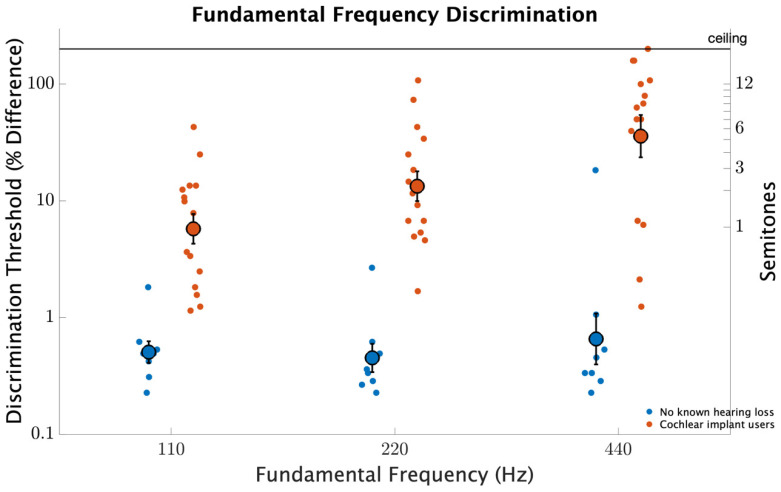
Fundamental frequency discrimination thresholds as a percentage difference measured for fundamental frequencies of 110, 220, and 440 Hz. The smaller circles represent individual thresholds. The larger circles with error bars represent participant averages with error bars indicating standard errors of the means.

**Figure 4 brainsci-12-00033-f004:**
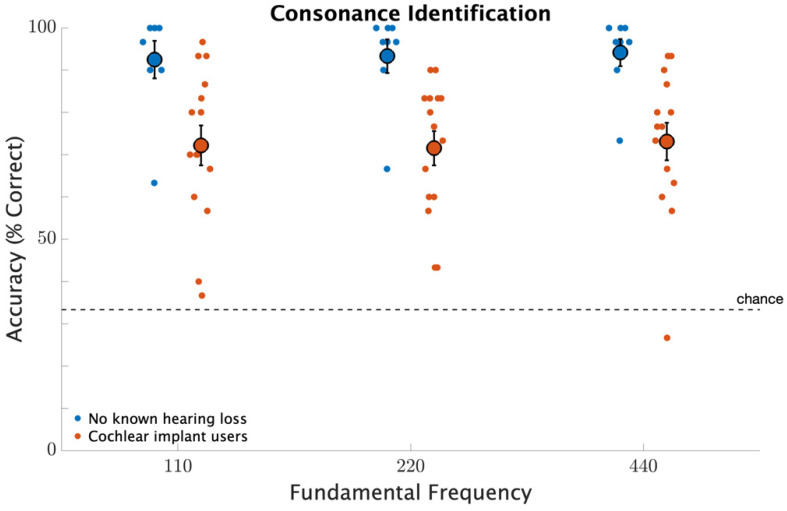
Consonance identification as a percent correct across distinct frequencies ranging from 110 to 440 Hz. The smaller circles represent individual identification measures. The larger circles represent participant averages with error bars indicating standard errors of the means.

**Figure 5 brainsci-12-00033-f005:**
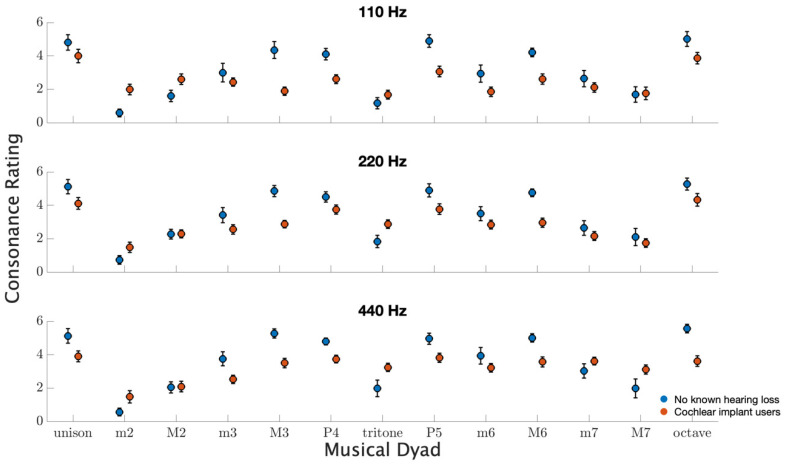
Pleasantness ratings on a scale from 0 to 6 for musical dyads from unison to octave. Each subplot indicates ratings for root notes near 110, 220, and 440 Hz. The circles represent the participant average with error bars indicating standard errors of the means.

**Figure 6 brainsci-12-00033-f006:**
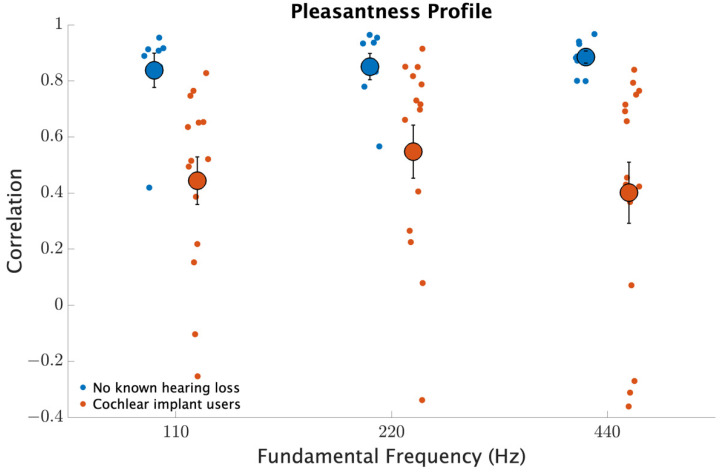
Correlation coefficient between individual pleasantness ratings and the average ratings from the group with no known hearing loss. The smaller circles represent individual correlations. The larger circles represent participant averages with error bars indicating the standard errors of the means.

**Figure 7 brainsci-12-00033-f007:**
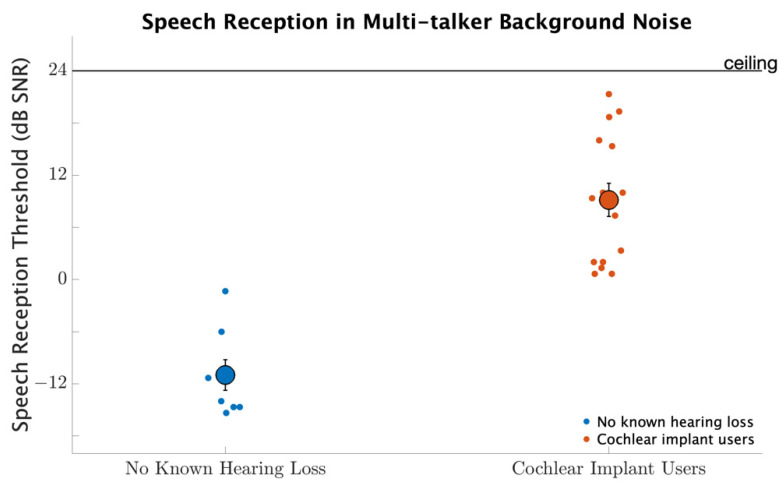
Speech reception thresholds for participants with no known hearing loss and for CI users. The smaller circles represent individual thresholds. The larger circle represents group averages with error bars indicating standard errors of the means.

**Figure 8 brainsci-12-00033-f008:**
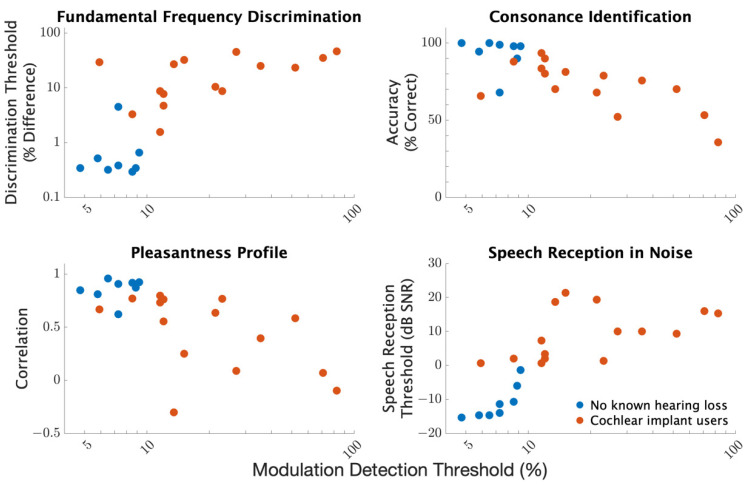
Comparisons of individual results from different procedures based on averages across conditions. For each comparison, circles represent the average measure for each individual participant averaged across conditions and repetitions.

**Figure 9 brainsci-12-00033-f009:**
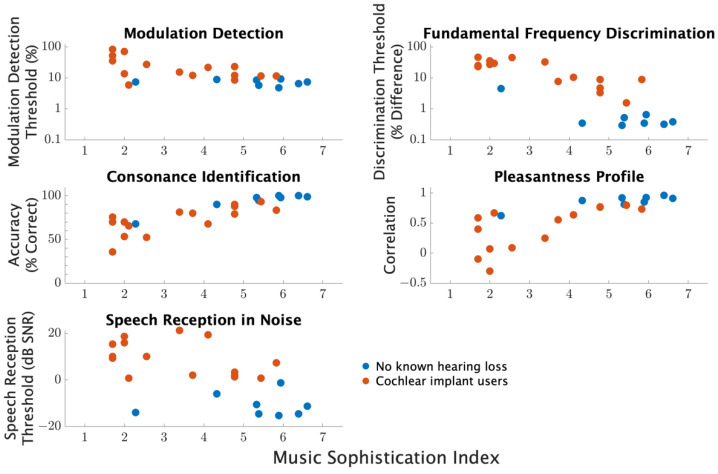
Comparisons of individual results from different procedures based on MSI scores. For each comparison, circles represent the average measure for each individual participant averaged across conditions and repetitions.

**Table 1 brainsci-12-00033-t001:** CI participant information. Age at time of testing and age at onset of hearing loss is given in years. Duration of profound hearing loss prior to implantation is given in years and estimated from subject interviews.

ID	Age	Gender	Etiology	Ear Tested	Age at Onset	Years Implanted	CI Company & Processor	Implant Model	Duration of Deafness	Method of Streaming
2	36	F	Unknown	Left/Right/Both Together	15	L:9 R:13	Cochlear N7s	L:CI24RE (CA) R:CI24RE (CA)	L:5 R:1	Mini Mic2
3	75	F	Progressive Nerve Loss	Left/Right/Both Together	40	L:21 R:17	Cochlear N6s	L:CI24R (CS) R:CI24RE (CA)	L:1 R:5	Cochlear Binaural Cable
5	83	M	Noise Induced	Right	40	13	Cochlear N6	CI24RE (CA)	20	Juster Multimedia Speaker SP-689
13	59	M	Mumps Disease	Right	14	3	Med-El Sonnet	Sonata	42	I-loop streaming
15	57	M	Ototoxic Medicine	Left	54	1	Advanced Bionics Naida	HiRes Ultra 3D CI with HiFocus Mid-Scala Electrode	1	Bluetooth/Compilot
17	74	F	Unknown	Left/Right/Both Together	Birth	L:20 R:15	Cochlear N6s	L:CI24R (CS) R:CI24RE (CA)	L:9 R:9	Free Field through HP Computer Speakers
18	72	F	Measles In Utero	Right	Birth	L:12 R:10	Cochlear N6s	L:CI24RE (CA) R:CI512	L:1 R:1	Computer Speakers
20	66	F	Unknown	Right	18	L:4 R:5	L:Cochlear N6 R:Cochlear N7	L:CI522 R:CI522	L:14 R:16	Free Field through iPad Speakers
25	68	M	Unknown	Right	44	4	Cochlear N6	CI552	20	Mini Mic2

**Table 2 brainsci-12-00033-t002:** Correlation coefficients comparing individual results from different procedures averaged across conditions. Only correlation magnitudes are displayed, but all correlation directions indicate better performance on one measure corresponding to better performance on the other measure unless marked with a †. Correlation analyses were performed across all participants and within each group. Abbreviations: modulation detection thresholds (MDT), fundamental frequency discrimination thresholds (F0DT), consonance identification (CID), pleasantness ratings (PR), speech reception thresholds (SRT), p<0.05 (*), p<0.01 (**), and p<0.001 (***).

All Participants		F0DT	CID	PR	SRT
	MDT	0.72 ^***^	0.76 ^***^	0.66 ^***^	0.73 ^***^
	F0DT		0.85 ^***^	0.79 ^***^	0.84 ^***^
	CID			0.80 ^***^	0.66 ^***^
	PR				0.75 ^***^
No Known Hearing Loss		F0DT	CID	PR	SRT
	MDT	0.09	0.16	0.18 ^†^	0.81 ^*^
	F0DT		0.92 ^**^	0.89 ^**^	0.09 ^†^
	CID			0.91 ^**^	0.12 ^†^
	PR				0.34 ^†^
CI Users		F0DT	CID	PR	SRT
	MDT	0.54 ^*^	0.69 ^**^	0.50	0.49
	F0DT		0.80 ^***^	0.75 ^***^	0.62 ^*^
	CID			0.70 ^**^	0.48
	PR				0.72 ^**^

**Table 3 brainsci-12-00033-t003:** Correlation coefficients comparing MSI to performance on perceptual measures. Only correlation magnitudes are displayed, but all correlation directions indicate higher MSI corresponding to better performance on the perceptual measure unless marked with a †. Abbreviations: modulation detection thresholds (MDT), fundamental frequency discrimination thresholds (F0DT), consonance identification (CID), pleasantness ratings (PR), speech reception thresholds (SRT), p<0.05 (*), p<0.01 (**), and p<0.001 (***).

		MDT	F0DT	CID	PR	SRT
All Participants	MSI	0.65 ^***^	0.83 ^***^	0.85 ^***^	0.77 ^***^	0.60 ^**^
No Known Hearing Loss	MSI	0.20	0.81 ^*^	0.96 ^***^	0.87 ^**^	0.06 ^†^
CI Users	MSI	0.56 ^*^	0.85 ^***^	0.72 ^**^	0.70 ^**^	0.45

## Data Availability

Main data generated and analyzed during this study are included in this article. Further enquiries can be directed to the corresponding author.

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
