# Peer review of "Pleasantness Ratings of Musical Dyads in Cochlear Implant Users"

_brainsci, 2021, doi:10.3390/brainsci12010033_

Round 1

Reviewer 1 Report

The manuscript is a well documented study of pleasantness ratings in cochlear implant users. I have just some minor comments:

- In Introduction, explanations (word definition) of specific words such as musical dyads, modulations, pitch, etc. are missing. They are needed for the readers to understand well.

- In Methods, explanations of the device Team Hearing is needed. What can it detect ?

- In Results, p values for pleasantness ratings are missing.

- Table 2,3,4 and Table 5,6,7 can be put together.

- In Discussion, the limitations of the study must be emphasized. Low number of cases, different implants and streaming used among the cases, etc. should all be mentioned. The clinical application of this study is also missing. How can we enhance pleasantness in CI patients ?

Author Response

Response to Reviewer

Thank you for the time and energy given to the review of our manuscript. We revised the manuscript as follows:

- In Introduction, explanations (word definition) of specific words such as musical dyads, modulations, pitch, etc. are missing. They are needed for the readers to understand well.

We revised the second paragraph to introduce modulation terminology as it relates to auditory physiology. We extended the third paragraph to further clarify the physiological terms. Timbre is defined in the fourth paragraph starting on line 78. The term “dyads” is defined around line 104.

- In Methods, explanations of the device TeamHearing is needed. What can it detect?

The following description has been added (line 175):

The TeamHearing web application includes a range of speech and pitch perception tests created to measure various aspects of hearing including musical judgements, psychophysical discriminations, and speech reception in various environments and noise conditions. The specific TeamHearing measures used for this study are described in the follow sections.

- In Results, p values for pleasantness ratings are missing.

Statistics have been added and the text revised (line 202).

- Table 2,3,4 and Table 5,6,7 can be put together.

The tables have been combined.

- In Discussion, the limitations of the study must be emphasized. Low number of cases, different implants and streaming used among the cases, etc. should all be mentioned. The clinical application of this study is also missing. How can we enhance pleasantness in CI patients?

The revision has the following discussion of limitations and clinical relevance (line 464):

The present study demonstrates the importance of sensitivity to modulation and pitch detection in combination with music training in order to enhance consonance and dissonance identification abilities among CI users. With the ability to discriminate between consonant and dissonant sounds in music, comes the potential to identify sounds that are more pleasant to an individual’s music listening [41]. In addition to these findings, there are certain limitation to consider. The limited number of participants and their individual differences should be considered. For example, some participants had years of musical training experience and had a greater understanding of consonance and the expected pleasantness of various musical intervals. Additionally, years of experience using cochlear implants, implant layout, and streaming method were not controlled among participants.

Reviewer 2 Report

In this manuscript, Manchala  et al., compared aspects of modulation and pitch perception in cochlear implant users and how it influenced the user's consonance perception and speech reception in noise. They also examined the effect of musical experience in these tasks. The authors have described the methodology in sufficient detail, the results are presented clearly with excellent statistical reporting. In my opinion, the authors conclusions are supported by the results presented. Overall this is a very well-written manuscript, I really have no critique.

Author Response

Thank you for reviewing our manuscript. We have made minor revisions based on one other reviews. As no specific revisions were requested, we thank you again for taking the time to consider our manuscript.